# Marine ecosystem role in setting up preindustrial and future climate

**Jerry F. Tjiputra** ✉, **Damien Couespel** **& Richard Sanders**

The ocean ecosystem is a vital component of the global carbon cycle, storing enough carbon to keep atmospheric $CO_2$ considerably lower than it would otherwise be. However, this conception is based on simple models, neglecting the coupled land-ocean feedback. Using an interactive Earth system model, we show that the role ocean biology plays in controlling atmospheric $CO_2$ is more complex than previously thought. Atmospheric $CO_2$ in a new equilibrium state after the biological pump is shut down increases by more than 50% (163 ppm), lower than expected as approximately half the carbon lost from the ocean is adsorbed by the land. The abiotic ocean is less capable of taking up anthropogenic carbon due to the warmer climate, an absent biological surface $pCO_2$ deficit and a higher Revelle factor. Prioritizing research on and preserving marine ecosystem functioning would be crucial to mitigate climate change and the risks associated with it.

As a key climate regulator, the ocean has slowed anthropogenic climate change by absorbing 91% of the heat trapped in the Earth system between 1971 and 2018[1] and 26% of anthropogenic carbon emitted since the preindustrial period[2]. Subsequently, it will play a crucial role in determining future climate. Air-sea fluxes of carbon dioxide primarily arise from the air-sea disequilibrium of the partial pressure of $CO_2$, driven by two main mechanisms: (i) the solubility pump, associated with $CO_2$ gas solubility and circulation, and (ii) the biological pump, the photosynthetic conversion of near-surface dissolved inorganic carbon (DIC) into organic matter that is exported into the ocean interior via gravitational sinking and circulation[3]. Through mechanism (ii), the biological carbon pump (BCP) effectively reduces surface-preformed DIC while increasing remineralized DIC at depth. This vertical redistribution and variations in preformed to remineralized DIC determines the partitioning of ocean-atmosphere carbon stocks[4]. Using an atmosphere-ocean equilibrium relationship, it is estimated that when the remineralized carbon stock is removed, atmospheric $CO_2$ levels would be approximately 150-240 ppm higher in the new equilibrium state[5].

Currently the BCP is often considered to operate at steady state, with the physically-driven solubility pump considered to have dominated the projected increase in current and future carbon sink and storage rates[6,7], due to the limited observational evidence and knowledge on the non-linear interactions and feedbacks between BCP and other Earth system components[8]. As a result, there is currently no consensus on the projected changes in BCP efficiency.

Repeated attempts to quantify the importance of BCP for future carbon storage have been conducted using ocean models of varying complexities. These include altering the chemical stoichiometry of organic matter production[9], changing the remineralization depth[10], enhancing micronutrient fertilization[11], and simulating future climate change with an abiotic ocean (i.e., by complete removal of all marine organisms)[12], among others. All these studies were performed with stand-alone ocean models, neglecting the carbon cycle feedback processes that occur between the ocean, land, and atmosphere. Such approaches can either underestimate or overestimate the impact of changing BCP on the long-term ocean carbon uptake[8]. In the face of rapidly intensifying climate change and anthropogenic pressures on marine ecosystems, better quantification of BCP's role in mitigating future climate change is of fundamental importance for climate science and policy developments.

Here, we assess the role of the BCP in setting up the preindustrial climate and quantify the impacts of its removal on future carbon sequestration and climate change. We apply the state-of-the-art Norwegian Earth System Model (NorESM2)[13,14] in its fully interactive configuration to simulate quasi-equilibrium preindustrial climate and subsequently project historical and future climate scenarios with and without marine organisms (see "Methods"). The impacts across three

NORCE Norwegian Research Centre AS, Bjerknes Centre for Climate Research, Bergen, Norway. ✉e-mail: jetj@norceresearch.no

future scenarios, ranging from high $CO_2$ emissions SSP5-8.5 (Shared Socioeconomic Pathways) to a strong mitigation SSP1-2.6[15], are investigated. Our simulations allow for prognostic atmospheric $CO_2$ concentration, taking into account carbon fluxes from land and ocean and the associated feedback to the climate system. The preindustrial reorganization of the carbon pools across the Earth system in the absence of BCP leads to more than 50% (163 ppm) higher atmospheric $CO_2$ and a 1.6 °C warmer surface temperature. Consequently, the Earth system's capacity to buffer anthropogenically-induced climate change is hampered. Despite having lower DIC content, an abiotic ocean counterintuitively has a higher surface DIC and $pCO_2$ concentration, leading to a higher Revelle factor and a reduced anthropogenic carbon uptake. Future land and ocean carbon sinks are projected to reduce considerably, accelerating and amplifying anthropogenic climate change.

## Results

### Preindustrial states

Two distinct quasi-equilibrium preindustrial climate states are simulated, with and without ocean biology (hereafter referred to as *REF* and *Abiotic*, respectively; see Methods and Supplementary Table 1). Key climate metrics and carbon budget estimates for these two preindustrial states are summarized in Table 1. The *Abiotic* ocean releases roughly 730 Pg C to the atmosphere during the 2000-year spin-up period (Supplementary Fig. 1). Of this, 345 Pg C is adsorbed by land, primarily through $CO_2$ fertilization-induced vegetation growth[16]. The remainder stays in the atmosphere, leading to a 163 ppm higher $CO_2$ concentration (445 ppm) compared to *REF* (282 ppm). This increases global mean surface temperature by 1.64 °C, while sea surface temperature (SST) increases by 1.15 °C. This warming leads to considerable impacts on the Earth system, such as reducing Atlantic Meridional Overturning Circulation (AMOC) strength (9%) and sea-ice area in the Arctic and Antarctic (23 and 24%, respectively; Table 1). Changes in sea-ice extent are seasonally non-uniform, with September Arctic sea-ice reducing the most, by more than half. Without ocean biology, global ocean temperature increases from 3.56 to 4.66 °C, and as the surface ocean equilibrates with the higher atmospheric $CO_2$ concentration, its Revelle factor increases by 16%.

The spatial patterns of surface temperature in the two preindustrial climate states are similar but warmer everywhere in the *Abiotic* (Supplementary Fig. 2a, b). The *Abiotic* ocean exhibits stronger $CO_2$ outgassing in the tropical oceans and ingassing in the mid-to-high latitudes (Supplementary Fig. 2d, e). The zonal outgassing band along the Southern Ocean circumpolar fronts seen in *REF*, associated with the upwelling of remineralized carbon-rich watermasses[17], switches into a net sink in *Abiotic*. In the subtropical oligotrophic, the rates of $CO_2$ flux are relatively similar in *REF* and *Abiotic*. The altered land carbon budget includes an increased vegetation carbon pool, with the largest increase simulated in the tropics region, followed by high-latitude ecosystems in the northern hemisphere (Supplementary Fig. 2g, h).

The stronger and expanded tropical outgassing in the *Abiotic* is consistent with (i) an absence of biological consumption of surface DIC and (ii) warming-induced lower $CO_2$ solubility, both of which increase surface $pCO_2$. In the high latitudes, stronger uptakes can be explained by the amplification of the seasonal cycle. Supplementary Fig. 3 illustrates that the removal of ocean biology fundamentally alters the seasonal cycle of the surface carbonate system. Firstly, the thermal-driven $pCO_2$ variability is amplified due to higher background $pCO_2$[18] and higher SST seasonal variation. Secondly, the lack of summer productivity and winter upwelling of remineralized carbon-rich deep water amplify the thermally driven $pCO_2$ variability. Thirdly, the indirect effect of land biosphere changes (stronger summer productivity and winter respiration) leads to an amplification of the atmospheric $CO_2$ seasonal cycle (Supplementary Fig. 3a,d). These three effects combine to increase the air-sea $pCO_2$ gradient, with a stronger effect during winter, and manifest in a stronger high-latitude carbon sink in the *Abiotic* ocean (with up to a three-fold increase; Supplementary Fig. 3c, f). We note that in our quasi-equilibrium preindustrial Abiotic ocean, the sedimentary carbon content is reduced by 391 Pg C (Table 1), due to the absence of organic matter accumulations and dissolution of organic materials. This additional DIC to the water column could contribute to the overall oceanic carbon release. We also note that our experiment represents an extreme hypothetical case with a complete absence of marine productivity. When we consider a 10% reduction in primary production, a range projected by models[19], Supplementary Fig. 4 shows that the preindustrial atmospheric $CO_2$ would approximately be 10 ppm higher, while the land carbon budget increases by 9 Pg C, implying ocean outgassing of approximately 30 Pg C.

### Historical and future projections

Projections of historical and future climate (1850–2100) show accelerated climate change in *Abiotic* relative to *REF* (Fig. 1 and Supplementary Fig. 5). Atmospheric $CO_2$ in *REF* grows from 281 to 415 (SSP1-

**Table 1 | Global mean and projected change in key climate metrics and carbon budgets in *REF* and *Abiotic* simulations**

| Variables | Preindustrial | | Δ (SSP5-8.5) | | |
| --- | --- | --- | --- | --- | --- |
| | Reference | Abiotic | Reference | Abiotic | Units |
| Surface air temperature | 14.52 | 16.16 | 3.77 | 4.88 | °C |
| Atmospheric $CO_2$ | 282 | 445 | 727 | 925 | ppm |
| Sea surface temperature | 18.78 | 19.93 | 2.48 | 3.48 | °C |
| Ocean temperature | 3.56 | 4.66 | 0.36 | 0.62 | °C |
| Maximum AMOC | 20.40 | 18.62 | −10.45 | −10.53 | Sv |
| Dissolved inorganic carbon | 37694 | 37367 | 497 | 397 | Pg C |
| Marine carbon sediment | 1853 | 1462 | 203 | −29 | Pg C |
| Sea-to-air $CO_2$ flux | − 0.11 | 0.07 | −5.45 | −3.59 | Pg C yr$^{-1}$ |
| Surface ocean Revelle factor | 9.83 | 11.40 | 3.21 | 2.97 | – |
| Land vegetation carbon | 537 | 812 | 202 | 96 | Pg C |
| Land soil and litter carbon | 2578 | 2646 | 71 | −41 | Pg C |
| Land-to-air $CO_2$ flux | 0.25 | − 0.02 | −5.84 | −3.26 | Pg C yr$^{-1}$ |
| Arctic sea-ice area | 11.15 | 8.57 | −6.22 | −8.12 | $10^6$ km$^2$ |
| Antarctic sea-ice area | 6.85 | 5.19 | −1.32 | −2.76 | $10^6$ km$^2$ |

Mean values are averaged over the first 10 years of the preindustrial control simulations, and projected change (Δ) are the difference between the 2091–2100 and 1851–1890 periods from the SSP5-8.5 future scenario and historical simulations.

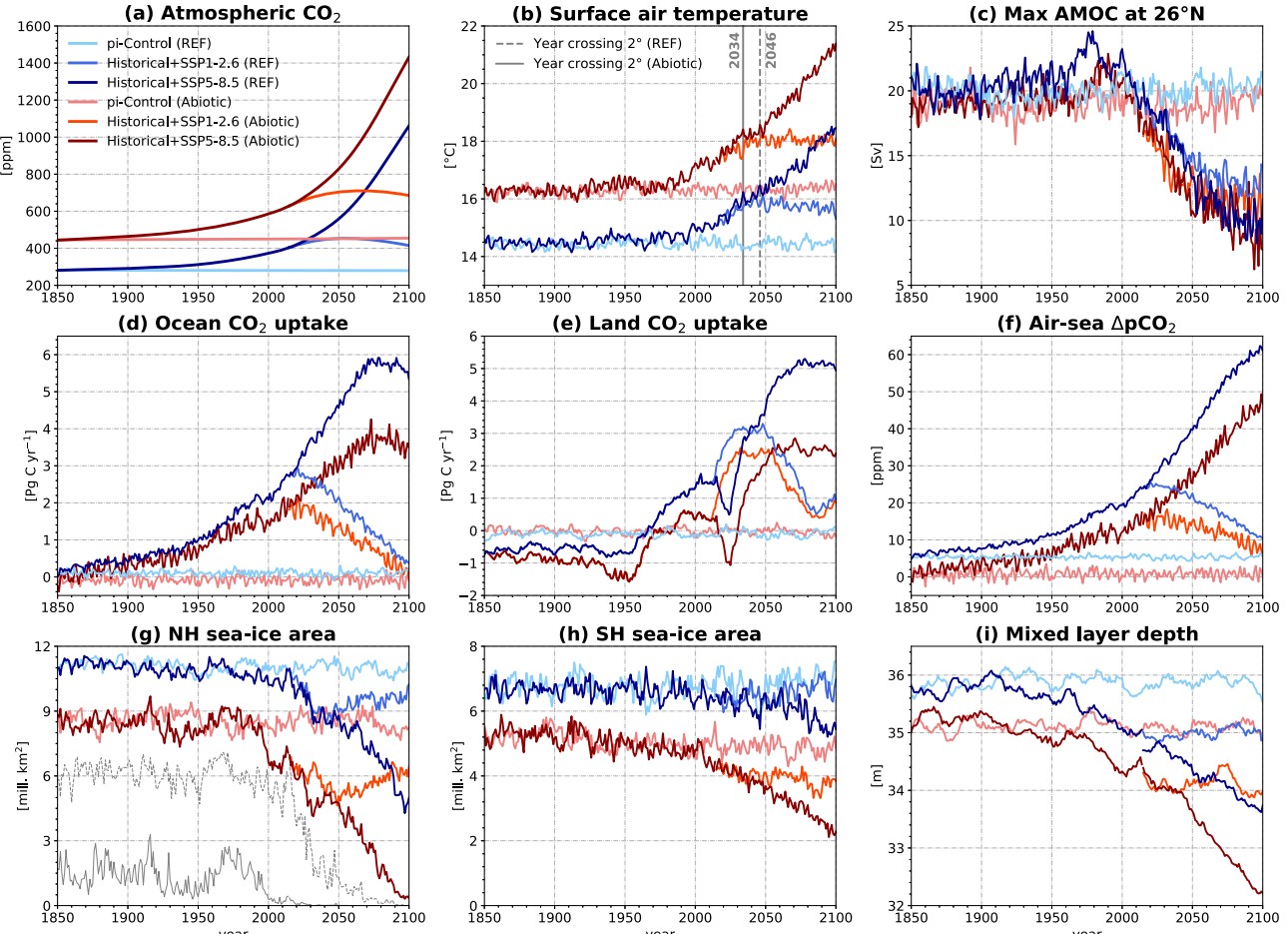

**Fig. 1 | Historical and future projections of global climate and carbon cycle states.** Time-series of the global annual mean (**a**) atmospheric $CO_2$ concentrations, (**b**) surface air temperatures, (**c**), Atlantic Meridional Overturning Circulation strengths, (**d**) ocean $CO_2$ uptakes, (**e**) 10-yr running mean of land $CO_2$ uptakes, (**f**) air-sea $\Delta pCO_2$, (**g**) northern hemisphere sea-ice area, (**h**) southern hemisphere sea-ice area, and (**i**) ocean mixed layer depths for *REF* (blue-lines) and *Abiotic* (red-lines) NorESM2-LM simulations under pre-industrial control, historical, SSP1-2.6 and SSP5-8.5 scenarios. Respective values for SSP2-4.5 are shown in Supplementary Fig. 5. Solid (dashed) gray lines in panel (**g**) depict the September sea-ice area in the *Abiotic* (*REF*) experiment under the historical and SSP5-8.5 scenarios.

2.6), 569 (SSP2-4.5), and 1061 (SSP5-8.5) ppm by 2100 (Fig. 1a and Supplementary Fig. 5a), consistent with the $CO_2$ pathways defined in the CMIP6 (Coupled Model Intercomparison Project phase 6) protocol[15]. In the *Abiotic*, $CO_2$ increases from 445 to 685 (SSP1-2.6), 883 (SSP2-4.5), and 1433 (SSP5-8.5) ppm. The stronger $CO_2$ increase in the *Abiotic* (by 106, 150, and 208 ppm in SSP1-2.6, SSP2-4.5, and SSP5-8.5, respectively) is caused by the land and ocean carbon sinks diminishing by as much as 91% (i.e., land sink in SSP2-4.5). The weaker ocean carbon sink, despite higher atmospheric $CO_2$, is attributed to the higher surface ocean $pCO_2$ and lower air-sea $pCO_2$ disequilibrium (Fig. 1f). Figure 2 summarizes that without ocean biology, 68 to 83% of the fossil fuel emissions would remain in the atmosphere by 2100, as compared to only 37 to 65% in simulations that include ocean biology.

These larger anthropogenic $CO_2$ residuals being stored in the atmosphere translate to a stronger climate sensitivity by accelerating climate change in *Abiotic*, with broad implications for various components of the Earth system (Fig. 1 and Supplementary Fig. 5). Under SSP5-8.5, the global mean surface temperature in *Abiotic* increases by nearly 5 °C by 2100, a 30% stronger warming rate than in *REF* (Table 1). Regionally, the largest warming occurs at high latitudes, with approximately 2 °C additional warming in the Arctic due to polar amplification[20] (Supplementary Fig. 6). Similarly, the AMOC strength, the sea-ice area, and the ocean mixed layer depth, are projected to decline faster in *Abiotic* (Fig. 1c, g–i). A complete removal of ocean biology leads to the system crossing the 2 °C warming threshold more

than ten years earlier, with the disappearance of summer Arctic sea ice occurring more than 50 years earlier (Fig. 1b, g). This accelerated climate change is also evident in the lower emissions scenarios of SSP1-2.6 and SSP2-4.5 (Supplementary Table 2). The transient climate response to $CO_2$ emissions (TCRE) for the *REF* and *Abiotic* are 1.69 and 2.13 °C Eg $C^{-1}$, respectively (Supplementary Fig. 7), implying a 20% reduction of the allowable $CO_2$ emissions for a specified global warming target when ocean biology is removed.

The reduced terrestrial carbon sinks in the *Abiotic* reflects the saturation of land carbon uptake in a warmer, higher $CO_2$ world due to the higher preindustrial vegetation carbon pool[21,22] (Fig. 1e and Table 1), hence weakening the negative terrestrial carbon cycle feedback associated with $CO_2$ fertilization. In the ocean, warmer and higher surface $pCO_2$ reduces seawater $CO_2$ solubility and buffering capacity[23], leading to a lower air-sea $pCO_2$ disequilibrium and less $CO_2$ uptake for a given increase in atmospheric $CO_2$ concentration (Fig. 1d, f and Table 1). The lack of ocean production-induced $pCO_2$ deficit, which has significant impacts at higher atmospheric $CO_2$ levels, also plays an important role[24].

Next, we analyze how the spatio-temporal dynamics of the anthropogenic carbon ($C_{ant}$) sink and storage in the ocean are altered by the absence of ocean biology. The cumulative ocean uptake of $C_{ant}$ (i.e., the difference between the transient historical + future and the preindustrial control simulations) over the 1850–2100 period is spatially heterogeneous, with deep water formation regions of the North

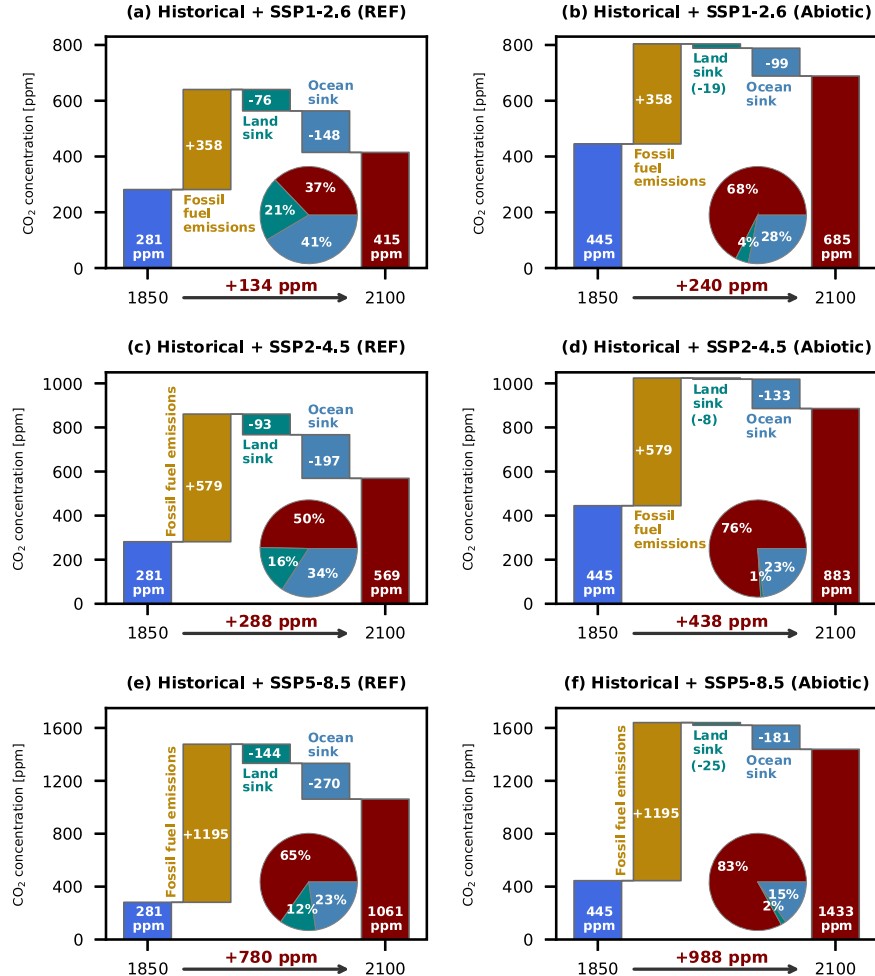

**Fig. 2 | Projected changes in the global carbon budget from 1850 to 2100.** Shown are the initial $CO_2$ concentration at the start of the historical period (dark blue), cumulative release of fossil fuel emissions (yellow), net land sink (including land-use changes, green), net ocean sink (light blue), and the final $CO_2$ concentration at the end of the 21st century (red) in [ppm] units. The pie charts depict fractions [in percentage] of fossil fuel emissions taken up by the land, ocean, and atmosphere. Values are from *REF* and *Abiotic* simulations for (**a**, **b**) historical + SSP1-2.6, (**c**, **d**) historical + SSP2-4.5, and (**e**, **f**) historical + SSP5-8.5 scenarios.

Atlantic and the Southern Ocean emerging as the most intense sink regions (Fig. 3). These key gateways for *Cant* sinks allow for efficient transport of *Cant*-rich surface water to the interior for long-term storage[25,26]. Stronger uptake is simulated in the high-emission SSP5-8.5 than SSP2-4.5 and SSP1-2.6, with regions of net *Cant* outgassing (more prominently in *Abiotic*) are simulated in the subtropics, where surface *Cant* converges[27]. The *Abiotic* Ocean absorbs considerably less *Cant* than *REF* in the subpolar North Atlantic and parts of the Southern Ocean (Fig. 3c, f, i).

We investigate the driving mechanisms of the lower *Abiotic Cant* uptake, focusing in the subpolar North Atlantic (dashed green outlines in Fig. 3c, f, i). Since the seasonal cycle of $CO_2$ fluxes is fundamentally altered in *Abiotic*, we assess the projection of different drivers in each season (Supplementary Fig. 8). The air-sea gradient of $pCO_2$ determines to first order the direction of $CO_2$ flux. In *REF*, the oceanic $pCO_2$ stays below the atmospheric values, and the subpolar North Atlantic is a net $CO_2$ sink all year, consistent with observations and models[28]. This is also seen in the *Abiotic* at the beginning of the historical period but is progressively altered by the faster oceanic $pCO_2$ growth rate, resulting in a remarkable decline in the $CO_2$ sink. More importantly, under SSP5-8.5 (and other scenarios) oceanic $pCO_2$ grows faster than atmospheric $pCO_2$ during spring and summer, therefore switching from uptake to outgassing. This is primarily driven by the combination of stronger warming and the higher initial surface Revelle factor, which reduce the

*Abiotic* ocean capacity to offset the growing atmospheric $pCO_2$ (Supplementary Fig. 9), as depicted by the weaker surface DIC growth rate (Supplementary Fig. 8).

## Discussion

Accurate assessment of the role of the marine ecosystem in regulating the Earth's climate is challenging due to the non-linear interactions and feedback that link different components of the Earth system. Applying a fully interactive Earth system model, we show that a complete removal of marine ecosystems increases preindustrial atmospheric $CO_2$ by 163 ppm, at the lower end of the range estimate using atmosphere-ocean equilibrium relationship (150–240 ppm)[5]. A previous study using a coupled physical-biogeochemical ocean model, and thus excluding the land feedback, gave a greater increase of atmospheric $CO_2$ (>200 ppm) after only 250 years[29]. In contrast, a theoretical study linking atmospheric $CO_2$ and the efficiency of BCP predicts only a 100 ppm increase when BCP is shut down[4]. In our simulations, the positive feedback due to climate warming as atmospheric $CO_2$ increases is overshadowed by negative feedback from the land biosphere, which absorbs nearly half of the released oceanic $CO_2$. Nevertheless, there are other feedbacks that are not considered, such as (i) reduction in the biogenic marine dimethyl sulfide emissions, which act as aerosol cooling agents[30], and (ii) warming-induced methane release from the permafrost[31]. These feedbacks would

# Ocean anthropogenic CO$_2$ uptake (1850-2100)

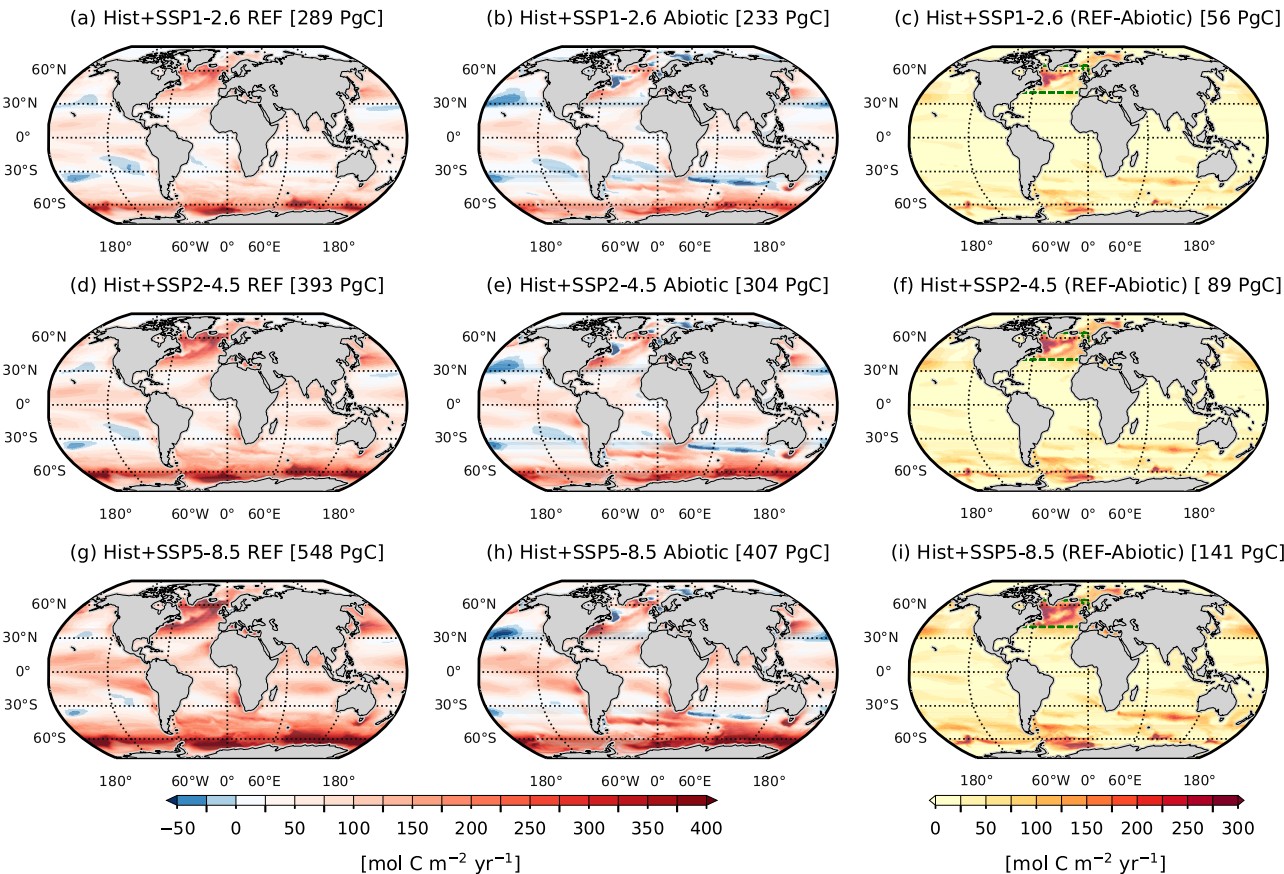

**Fig. 3 | Maps of ocean uptake of anthropogenic carbon.** Spatial patterns of cumulative (1850–2100) anthropogenic carbon uptake by the ocean under (**a**, **b**) historical + SSP1-2.6, (**d**, **e**) historical + SSP2-4.5, and (**g**, **h**) historical + SSP5-8.5 scenarios for *REF* and *Abiotic* simulations. Panels (**c**, **f**, **i**) depict differences between *REF* and *Abiotic* simulations.

enhance warming-induced climate feedback, further increasing the preindustrial atmospheric CO$_2$ level. Therefore, our estimated climate impacts likely represent the lower end of the expected range.

The reorganization of carbon pools within the Earth system alters the trajectories of anthropogenic climate change. The newly equilibrated abiotic ocean is warmer and contains less DIC. The absence of a biologically mediated vertical DIC gradient (Supplementary Fig. 10) leads to a higher surface pCO$_2$ and Revelle factor. This higher surface Revelle factor impedes the oceanic sink of anthropogenic CO$_2$[23], implying that future changes in primary production may negatively affect anthropogenic CO$_2$ uptake. Similarly, the reallocation of carbon from the ocean to the land biosphere saturates the vegetation growth response to higher atmospheric CO$_2$[21]. Depending on the emissions scenario, the ocean and land CO$_2$ uptake in *Abiotic* could be reduced by up to 34% and 91%, respectively, increasing the cumulative airborne fraction of CO$_2$ emissions from 37%, 50%, 65% to 68%, 76%, 83% for the SSP1-2.6, SSP2-4.5, SSP5-8.5 scenarios (Fig. 2). The projected climate change rate is therefore enhanced, particularly in the polar regions. Considering the lack of the above-mentioned feedback processes, these climate change figures are likely underestimated. This accelerated climate change also illustrates the potential consequence if the longevity of natural carbon sink is reduced faster or earlier than initially expected[32].

The simulated changes shown here represent extreme end-member examples of the role that ocean biology plays in controlling atmospheric CO$_2$, both in the steady state and under greenhouse forcing. However, they are valuable and clearly show that estimates of the effects of biologically-mediated ocean CO$_2$ storage in the pre-industrial has on atmospheric CO$_2$ based on simple metrics, i.e., integrated remineralised CO$_2$, need to be treated with some cautions as any reduction in this term would likely lead to an enhanced terrestrial carbon sink. In addition, our simulations indicate that ocean biological processes play a crucial role in driving the uptake of *Cant* in the modern ocean by adjusting the spatial and temporal patterns in the surface pCO$_2$ deficit.

Our assessment of *Cant* sequestration demonstrates that the largest impact of an abiotic ocean occurs in the ventilation regions of the North Atlantic and Southern Ocean, where long-term surface-to-interior *Cant* export is significantly reduced, particularly during summer, when the absence of biological export production and enhanced surface warming keep surface pCO$_2$ high (see Supplementary Fig. 9) and impede the ocean carbon sinks in the more stratified future ocean. Despite community consensus on the importance of ocean biology in the Earth system, its representations in state-of-the-art models are far from perfect. This leads to large uncertainty when simulating the observed CO$_2$ flux variability[33] and projecting future changes in ocean biology[7,19,34]. In Earth system models, the BCP is generally positively associated with surface primary productivity, nevertheless there is a nuanced relationship between primary production, carbon export, and carbon sequestration. Spatial and temporal variations in surface productivity and ecosystem structure may have non-linear impacts on BCP efficiency, complicating the direct links between reductions in

productivity and $CO_2$ sequestration[35]. Addressing this complexity would be valuable to guide future climate modeling efforts, carbon cycle research, and policy development. Interdisciplinary research that integrates marine biological observations and models using various approaches to constrain the impacts of future global change should also be embraced[36]. Our finding also contradicts the current notion that the biological carbon pump plays little to no role in adsorbing excess Cant from the atmosphere. Instead, it clearly plays an important role by setting up the disequilibria and Revelle factor conditions that determine the magnitude of this uptake.

In order to optimize the service the ocean offers us by mitigating anthropogenic climate change, our results emphasize the importance of sustaining a healthy and well-functioning marine ecosystem. Our simulations, though extreme, suggest that the dissolution of sedimentary organic material can alter the water column DIC budget and, subsequently, the air-sea $CO_2$ fluxes. This has obvious parallels to human-induced disturbances of the seafloor by trawling or dredging; however, these will be much smaller in size, and hence future studies applying a more realistic scenario would be necessary to investigate the impact of human-induced disruption of marine sediments.

In addition to improvements in various dynamical processes in the Earth system (cloud feedback, land carbon cycle, ocean circulation, albedo feedback, etc.), better representations of marine primary production and biological carbon pump should not be overlooked to constrain estimates of climate sensitivity and future climate projections. Future developments in ocean biogeochemistry models should emphasize improving the spatio-temporal biological processes in key ocean carbon sink regions (i.e., the North Atlantic and Southern Ocean). Sustained monitoring of marine biological carbon pumps would be necessary to improve our estimates of future ocean carbon sinks and, subsequently, the development of robust strategies for climate mitigation.

## Methods

### Model description and experimental design
We used the second generation of the Norwegian Earth System Model (NorESM2-LM), which contributed to the Coupled Model Intercomparison Project phase 6 (CMIP6)[37] and the sixth assessment report of the Intergovernmental Panel for Climate Change[38]. It couples atmosphere, ocean, sea-ice, and land modules and simulates the physical and biogeochemical interactions between them. Here the NorESM2-LM was configured in a fully interactive mode, allowing for prognostic atmospheric $CO_2$ concentrations by accounting for online air-sea and air-land $CO_2$ fluxes. A full description of the model components and their performance has been extensively validated and documented[13,14]. The NorESM2-LM simulates well the observed large scale pattern of surface primary productivity, and its annual rate is well within the range of observational estimates and other CMIP6 models[39]. Two sets of preindustrial climate state and future projections were performed: (i) reference (REF), where the ocean biology was not modified, and (ii) Abiotic, where the ocean primary productivity was switched off, and the ocean carbon sources and sinks were driven only by physical processes. The riverine fluxes of biogeochemical substances were deactivated in Abiotic to balance the diminished sinks of materials into the sediments. We also applied climatological marine dimethyl sulfide (DMS) fluxes based on observations to compensate for the radiative imbalance at the top of the atmosphere. This last step was necessary since the impact of diminishing phytoplankton-produced DMS on the climate system is currently not well understood.

Prior to transient historical and future simulations, the NorESM2-LM was spun up until a quasi-equilibrium preindustrial climate and carbon cycle states were achieved. For REF, the model was initialized from observations and spun up for 1600 years with a prescribed constant atmospheric $CO_2$ concentration of 284 ppm, followed by an additional spin-up with prognostic $CO_2$ for 250 years. For Abiotic, the

spin-up was started at the end of REF's spin-up, with the marine productivity module deactivated and integrated for 2000 years. Following these spin-ups, both REF and Abiotic reach a stable preindustrial climate state with sufficiently low drifts for the purpose of this study (Supplementary Table 1).

Here, we summarize the transition state occurring in the Abiotic spin-up. Following a complete cessation of marine productivity, the ocean starts to outgass carbon to the atmosphere, leading to a rapid decrease in DIC storage and an increase in atmospheric $CO_2$ in the first few hundred years (Supplementary Fig. 1). The lower DIC content is reflected by declining remineralized DIC and is compensated by the gradual increase in preformed DIC and the dissolution of sedimentary carbon. The resulting higher atmospheric $CO_2$ induces higher terrestrial primary production and vegetation growth through the $CO_2$ fertilization effect[16], removing a substantial fraction of the ocean carbon release. The soil and litter carbon pools also increase, though only slightly.

Starting from the two quasi-equilibrium preindustrial states, we performed transient historical (1850–2014) and three future scenarios (2015–2100) simulations following the CMIP6 protocol[37]. The three future scenarios considered were: (i) low $CO_2$ emissions SSP1-2.6 (Shared Socioeconomic Pathways), which is consistent with a 2 °C warming by 2100 relative to the preindustrial, (ii) moderate $CO_2$ emissions SSP2-4.5, and (iii) high $CO_2$ emissions SSP5-8.5. Corresponding preindustrial control simulations (1850-2100) for each REF and Abiotic were also performed. These preindustrial control simulations were applied to correct model drifts and estimate the anthropogenic carbon (Cant) content in the ocean. At the start of the transient historical simulation (1850), the total carbon inventories for the atmosphere, land, and ocean (water column+sediment) reservoirs for REF (Abiotic) are 599 (945), 3115 (3458), and 39547 (38829) Pg C, respectively. We note that the noticeable decline in net land $CO_2$ uptake in the early 21st century under the SSP5-8.5 scenario (Fig. 1e) is associated with the loss of carbon due to the prescribed land use change and fires.

### Uncertainty analysis
The results of our experiments, specifically on how the atmospheric $CO_2$ evolves with the release of oceanic carbon, depend on the land carbon cycle response or feedback to higher atmospheric $CO_2$. In our experiment the atmospheric $CO_2$ slowly increases from preindustrial level to 445 ppm in 2000 years. The most comparable experiment under the CMIP6 framework is the extended historical and SSP1-2.6 scenario (1850-2300), with the increasing atmospheric $CO_2$ stabilizes toward 396 ppm after 450 years of integration[40]. Five ESMs have provided their outputs: ACCESS-ESM1.5, CanESM5, IPSL-CM6-LR, MIROC-ES2L, and UKESM1-0-LL. Supplementary Fig. 11 shows that the multi-model mean of cumulative land carbon sink plus total land-use-related carbon budget (from year 1850–2150 is 215 Pg C)[41] amounts to 337 ± 26.73 Pg C, which is in good agreement with our total land carbon tbudget increase of 343 Pg C (Table 1) in the preindustrial Abiotic. We also assess the robustness of our projected preindustrial key climate response. Our simulated temperature increase, AMOC decline, and sea-ice cover decline are all within the range of CMIP6 ESMs (Supplementary Fig. 11). We note that under historical and SSP1-2.6 scenarios, in addition to atmospheric $CO_2$ increase, there are also aerosols and other prescribed forcings, such as volcanic eruption and other greenhouse gases, that could affect the climate.

## Data availability
The NorESM2-LM model outputs (Reference and Abiotic) for the historical and future scenarios simulations can be obtained at the Earth System Grid Federation portal https://esgf-node.ipsl.upmc.fr and https://doi.org/10.11582/2024.00083.

## Code availability

Model codes for the NorESMs model are publicly available in https://noresm-docs.readthedocs.io/en/noresm2/access/access.html.

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

## Acknowledgements

This work was funded by the European Union under grant agreement no. 101083922 (OceanICU, J.T., D.S., and R.S.) and UK Research and Innovation (UKRI) under the UK government's Horizon Europe funding guarantee [grant number 10054454, 10063673, 10064020, 10059241, 10079684, 10059012, 10048179]. The views, opinions, and practices used to produce this dataset/software are, however, those of the author(s) only and do not necessarily reflect those of the European Union or European Research Executive Agency. Neither the European Union nor the granting authority can be held responsible for them. This study is a contribution to the Research Council of Norway projects INES2 (no. 350390) and Navigate (no. 352142). We acknowledge Timothée Bourgeois for preparing the SSP2-4.5 $CO_2$ emissions file. High-performance computing and storage resources were provided by the Norwegian Research Infrastructure Services (projects nn1002k, nn2345k, and ns1002k).

## Author contributions

Funding acquisition: R.S. and J.F.T.; Conceptualization: J.F.T.; Analysis of the results: J.F.T., D.C., and R.S.; Writing: J.F.T., D.C., and R.S.; Editing: J.F.T., D.C., and R.S.

## Funding

## Competing interests

The authors declare no competing interests.
