## [Peer Review file · Nature Communications]

Marine ecosystem role in setting up preindustrial and future climate

Corresponding Author: Dr Jerry Tjiputra

Version 0:

Reviewer comments:

Reviewer #1

(Remarks to the Author)

The authors have fully addressed my concerns and I find the revised paper presents important findings suitable for publication in Nature Communications. In particular, the paper highlights and quantifies the response of the land sink to changes in the ocean, and the role of ecosystems in setting up mean pre-conditions that also have an influence on anthropogenic uptake of CO₂ by the oceans.

The authors have revised all the sentences which I identified as not being fully justified by the manuscript results (bar one, see below, which I had not originally seen). They have also introduced a more realistic simulation with a reduction of 10% in productivity which puts the full abiotic results into context.

I have a few comments remaining:

p. 6, lines 237-241. The comment on the role of ocean sediments appears in the discussion only and is not backed by analysis in the results section, other than a line that is not commented in Table 1. Given the strong comment made on the possible implications for deep-sea mining and trawling, the sentence on lines 240-241 needs to be backed by analysis in the results section. Here also, please be conscious that the abiotic results are extreme scenarios and should not be presented in a way that can be interpreted as plausible effects of mining/trawling, which would be considerably smaller and potentially negligible. The sentence rightly says this needs further investigation.

p. 7, lines 277-278. The study cited in reference 36 and the Guardian newspaper article cited in the letter to the editor grossly overinterpret the effect of recent climate variability in 2023 (linked to El Niño) as suggestion that it might indicate a collapse of the land sink. Study 36 simply adds Land sink + land-use change emissions (both affected by El Niño) to highlight a very low land sink. However, the land sink did not behave unusually compared to other El Niño periods. Reference 36 is a preprint that has yet to be challenged by peer-reviewed. Whereas the comment in the current paper is valid (that the longevity of the natural carbon sinks is not guaranteed), the reference is not, at least not by itself. For example, Friedlingstein et al. 2024 (also a preprint) makes the case that the land sink will recover in 2024. Please expand references here to back the statement made with peer-reviewed publications (such as the IPCC).

<https://essd.copernicus.org/preprints/essd-2024-519/>

Figure 1e. The land panel has discontinuity around 2020, both with the broken lines and with the following unusual and strong decrease in the land sink (around 2020-2040), which are not explained in the paper and appear wrong to me. These need to be verified and explained, at the minimum in the methods or Supplementary material.

On a side note to the authors, a similar analysis of an abiotic land would be interesting and complementary to the current paper.

Reviewer #2

(Remarks to the Author)

Reviewer #3

(Remarks to the Author)

Manuscript Review for Nature Communications

Submitted: 12/23/2025

Title: Marine ecosystem role in setting up preindustrial and future climate

The manuscript examines the biological carbon pump's (BCP) total role in the global ocean carbon sink and its importance in regulating historic and future atmospheric CO₂ and global temperature. The conclusions challenge the notion that the BCP plays a secondary role to the solubility and circulation components of the ocean sink in absorbing excess atmospheric CO₂, demonstrating how its absence increases the Revelle factor and surface pCO₂ under preindustrial conditions. The paper explores the carbon cycle's sensitivity to biological processes by examining its extreme boundaries, providing clear, conceptual comparisons. While the finding that disabling the BCP could cause climatologically significant atmospheric CO₂ and global temperature increases underscores its role in climate regulation, a total biological shut-down is not very plausible and represents an extreme case. In response to reviewer feedback, the authors incorporated a partial productivity reduction experiment for more realistic insights. By including the moderate case of (90%) productivity, consistent with the 10% decline projected by CMIP5 and CMIP6 models, the study expands its relevance.

The authors should consider providing the total inventories of carbon in each reservoir (land, ocean, atmosphere, wherever carbon is stored) at the beginning of their model runs. While it is true that similar studies often omit this information, it would be greatly beneficial for other modeling groups that want to compare the behavior of different Earth System models. Because ESMs are typically spun up with a prescribed atmospheric pCO₂, the inventory of carbon will be different in every model. As a result, the relative redistribution of carbon between the different reservoirs as a result of a change in the biological pump will be different for different models. If the reference model had a very weak BCP, then shutting it off would have a weaker impact. This applies to the land biosphere as well— the strength of the CO₂ fertilization effect, for example, would differ based on how the model is tuned.

The study makes several unique contributions that are significant to the field and related fields.

One strength of the study is its integration of the coupled land-ocean-atmosphere system, a relatively uncommon approach in BCP research. In the Main section, another study is mentioned that simulates future climate change with an abiotic ocean— however, that study used a stand-alone ocean model. Their interconnected framework adds important nuance to the field by including carbon cycle feedback processes that occur between land, ocean, and atmosphere.

In the discussion, the highlighted seasonality-driven ocean-atmosphere CO₂ dynamics, particularly in the North Atlantic and Southern Ocean, offer valuable insights into underexplored climate feedbacks related to seasonality. These feedbacks are particularly uncertain because of the lack of year-round temporal coverage of observed BCP variables. The amplification of seasonal pCO₂ outgassing in key ocean sink regions and its impact on long-term carbon sequestration is particularly noteworthy.

The study highlights noteworthy spatial differences in the storage of anthropogenic carbon between biotic and abiotic simulations, particularly within specific ocean regions such as the subpolar Atlantic. The authors also identify the mechanisms driving these differences with a well-supported explanation, demonstrating how factors like the faster oceanic pCO₂ growth rate, higher initial surface Revelle factor, and seasonal variations in air-sea pCO₂ gradients contribute to the reduced carbon uptake in abiotic simulations.

A few concerns could be addressed and do not prohibit publication.

The decision to include only high- and low- end emissions scenarios raises questions about the exclusion of moderate scenarios, which could yield unique feedback patterns beyond averaging the extremes. Including a brief discussion of this

choice would provide helpful context to the design of the study.

Key results of the paper rely on a direct link between increased/decreased primary productivity and increased/decreased carbon sequestration by the BCP. Increased primary productivity does not always lead to increases in effective carbon storage or even carbon export past shallow ocean layers. An acknowledgement of the nuanced relationship between primary productivity, carbon export, and sequestration in the discussion would be valuable. Variations in productivity by region or organism type may have differing impacts on BCP efficiency, complicating direct links between productivity reductions and CO₂ sequestration losses. Addressing this complexity would help the study in guiding future climate modeling efforts, carbon cycle research, and policy development.

Several points in the main text could be clarified to strengthen the narrative.

Lines 38-40: The explanation of how the BCP influences the ratio of preformed to remineralized carbon, and its direct link to total ocean carbon storage, could be clearer. Consider stating directly that the BCP reduces surface preformed carbon by converting it into export production carbon, thus increasing remineralized carbon at depth. A clarification would help avoid ambiguity, emphasizing that the BCP primarily redistributes carbon within the ocean rather than storing it in an isolated reservoir. Consider rephrasing to clarify and emphasize the specific role of the BCP in partitioning ocean-atmosphere carbon stocks.

Lines 40-42: The reference to “removing remineralized carbon” was initially confusing without clearer context. Since remineralized carbon results directly from BCP activity, specifying whether the term refers to modeled carbon stocks or the conceptual absence of the BCP would reduce potential misunderstandings. This could be interpreted as the authors taking out remineralized carbon, so that the deep ocean has less carbon, and then re-equilibrating the model. Alternatively, this could mean that the BCP was “shut off”, and then they got a readjustment in a different way with a different amount of carbon.

Lines 43-45: Consider clarifying in particular what aspects of the carbon sink are considered to be dominated by the solubility pump rather than BCP— for example, its current magnitude and variability, or projected changes under climate scenarios?

Lines 45-48: Consider mentioning the critical role of limited observational evidence and lack of consensus on current changes in BCP efficiency when discussing gaps in the understanding of the current and future BCP.

Lines 50-53: While the examples of modeling approaches used to study the BCP here are relevant, the list appears selective. Consider rephrasing to signal the listed examples are representative rather than exhaustive.

Lines 53-56: The assertion that neglecting feedback processes leads to overestimation of the BCP’s impact could benefit from a short explanation. Why does the omission consistently result in overestimation, rather than introducing more general uncertainty?

Lines 60-61: The term “marine ecosystem” might be too broad. If the study specifically shuts down the BCP, consider using that term to align more closely with the experiment design.

Version 1:

Reviewer comments:

Reviewer #2

(Remarks to the Author)

Reviewer #3

(Remarks to the Author)

We believe that the authors have addressed all the issues from the reviews.

Response letter to manuscript

Changes in response to Reviewers' comments are marked with 'green' color in the revised manuscript with track changes. Line numbers refer to the revised manuscript with track changes.

Reviewer #1 (Remarks to the Author):

The authors have fully addressed my concerns and I find the revised paper presents important findings suitable for publication in Nature Communications. In particular, the paper highlights and quantifies the response of the land sink to changes in the ocean, and the role of ecosystems in setting up mean pre-conditions that also have an influence on anthropogenic uptake of CO₂ by the oceans.

We thank reviewer#1 for his/her positive feedback and additional constructive comments. We have considered all the new comments in our latest revision.

The authors have revised all the sentences which I identified as not being fully justified by the manuscript results (bar one, see below, which I had not originally seen). They have also introduced a more realistic simulation with a reduction of 10% in productivity which puts the full abiotic results into context.

Responses to additional comments are provided below.

I have a few comments remaining:

p. 6, lines 237-241. The comment on the role of ocean sediments appears in the discussion only and is not backed by analysis in the results section, other than a line that is not commented in Table 1. Given the strong comment made on the possible implications for deep-sea mining and trawling, the sentence on lines 240-241 needs to be backed by analysis in the results section. Here also, please be conscious that the abiotic results are extreme scenarios and should not be presented in a way that can be interpreted as plausible effects of mining/trawling, which would be considerably smaller and potentially negligible. The sentence rightly says this needs further investigation.

In the revision (end of Sect. 'Preindustrial states'), we have commented on the role of sedimentary carbon loss (in Abiotic ocean) in increasing water column DIC and potentially the ocean outgassing:

L150-153: "We note that in our quasi-equilibrium preindustrial Abiotic ocean, the sedimentary carbon content is reduced by 391 Pg C (Table 1), due to the absence of organic matter accumulations and dissolution of organic materials. This additional DIC to the water column could contribute to the overall oceanic carbon release."

We agree that our extreme scenarios should not be interpreted as plausible effects of mining/trawling, and have rephrased the aforementioned text in the discussions from

“Our simulations also suggest that the dissolution of organic material in the ocean sediment considerably alters the water column DIC budget and subsequently the air-sea CO₂ fluxes. This implies that the impact of human-induced disruption of marine sediment, e.g., through deep sea mining or fish trawling, on the ocean carbon cycle should be thoroughly investigated.”

to:

L294-300: “Our simulations, though extreme, suggest that the dissolution of sedimentary organic material can alters the water column DIC budget and subsequently the air-sea CO₂ fluxes. This has obvious parallels to human induced disturbances of the seafloor by trawling or dredging; however, these will be much smaller in size and hence future studies applying a more realistic scenario would be necessary to investigate the impact of human-induced disruption of marine sediments.”

p. 7, lines 277-278. The study cited in reference 36 and the Gardian newspaper article cited in the letter to the editor grossly overinterpret the effect of recent climate variability in 2023 (linked to El Nino) as suggestion that it might indicate a collapse of the land sink. Study 36 simply adds Land sink + land-use change emissions (both affected by El Nino) to highlight a very low land sink. However, the land sink did not behave unusually compared to other El Nino periods. Reference 36 is a pre-print that has yet to be challenged by peer-reviewed. Whereas the comment in the current paper is valid (that the longevity of the natural carbon sinks is not guaranteed), the reference is not, at least not by itself. For example, Friedlingstein et al. 2024 (also a preprint) makes the case that the land sink will recover in 2024. Please expand references here to back the statement made with peer-reviewed publications (such as the IPCC). <https://essd.copernicus.org/preprints/essd-2024-519/>

We agree that reference#36 is not ideal when discussing to the longevity of natural carbon sinks. We have now rephrased the statement below and reference Ch. 5 of the IPCC-AR6 report (Canadell et al., 2021), which states the slowing down of natural CO₂ sink rate under future high-warming scenarios (i.e. box FAQ5.1):

“The projected lower land and ocean carbon sinks in the future provides a glimpse of potential accelerated climate change, in light of a recent study raising concern on the longevity of natural carbon sink [36].”

To (now moved to Discussion section):

L258-260: “This accelerated climate change also illustrates the potential consequence if the longevity of natural carbon sink is reduced faster or earlier than initially expected (Canadell et al., 2021).”

Figure 1e. The land panel has discontinuity around 2020, both with the broken lines and with the following unusual and strong decrease in the land sink (around 2020-2040), which are not explained in the paper and appear wrong to me. These need to be verified

and explained, at the minimum in the methods or Supplementary material.

Thank you for noticing this. The discontinuity is due to the separation between historical (1850-2014) and future scenario (2015-2100) 10-yr averaging applied to land carbon uptake. We have revised Fig. 1 in the revision. The strong decrease in 'net' land sink around 2020-2040 seen in SSP5-8.5 is associated with the carbon lost due to land-use change, fire and harvested carbon. Figure R1 below illustrates that the strong sink reduction disappears when the land carbon sink is calculated only based on net primary production minus respiration (NEP, panel c).

Figure R1. Time-series of 10-yr averaged of (a) net land carbon sink, (b) net ecosystem exchange (exclude land-use change), and (c) net ecosystem production (exclude land-use change, fires, and harvest pool fluxes) under the (dark-blue) historical and SSP5-85 scenario, and (light-blue) preindustrial control.

We have added the following at the end of Methods section:

L411-413: “We note that the noticeable decline in net land CO₂ uptake in the early 21st century under the SSP5-8.5 scenario (Fig. 1e) is associated with the loss of carbon due to the prescribed land use change and fires.”

On a side note to the authors, a similar analysis of an abiotic land would be interesting and complementary to the current paper.

We agree, though since land uptakes are primarily biotic-driven, this would consider only land use change.

Reviewer #2 (Remarks to the Author):

We thank reviewer#2 for supporting the co-review process of our manuscript.

Reviewer #3 (Remarks to the Author):

The manuscript examines the biological carbon pump's (BCP) total role in the global ocean carbon sink and its importance in regulating historic and future atmospheric CO₂ and global temperature. The conclusions challenge the notion that the BCP plays a secondary role to the solubility and circulation components of the ocean sink in absorbing excess atmospheric CO₂, demonstrating how its absence increases the Revelle factor and surface pCO₂ under preindustrial conditions. The paper explores the carbon cycle's sensitivity to biological processes by examining its extreme boundaries, providing clear, conceptual comparisons. While the finding that disabling the BCP could cause climatologically significant atmospheric CO₂ and global temperature increases underscores its role in climate regulation, a total biological shut-down is not very plausible and represents an extreme case. In response to reviewer feedback, the authors incorporated a partial productivity reduction experiment for more realistic insights. By including the moderate case of (90%) productivity, consistent with the 10% decline projected by CMIP5 and CMIP6 models, the study expands its relevance.

We thank reviewer #3 for providing thorough critical and constructive feedback on our manuscript. We have addressed all of the raised comments below.

The authors should consider providing the total inventories of carbon in each reservoir (land, ocean, atmosphere, wherever carbon is stored) at the beginning of their model runs. While it is true that similar studies often omit this information, it would be greatly beneficial for other modeling groups that want to compare the behavior of different Earth System models. Because ESMs are typically spun up with a prescribed atmospheric pCO₂, the inventory of carbon will be different in every model. As a result, the relative redistribution of carbon between the different reservoirs as a result of a change in the biological pump will be different for different models. If the reference model had a very weak BCP, then shutting it off would have a weaker impact. This applies to the land biosphere as well– the strength of the CO₂ fertilization effect, for example, would differ based on how the model is tuned.

As requested, we have added the following at the end of the 'Methods section':

L408-411: "At the start of the transient historical simulation (1850), the total carbon inventories for the atmosphere, land, and ocean (water column+sediment) reservoirs for REF (Abiotic) are 599 (945), 3115 (3458), and 39547 (38829) Pg C, respectively."

The study makes several unique contributions that are significant to the field and related fields. One strength of the study is its integration of the coupled land-ocean-atmosphere system, a relatively uncommon approach in BCP research. In the Main section, another study is mentioned that simulates future climate change with an abiotic ocean– however, that study used a stand-alone ocean model. Their interconnected framework adds important nuance to the field by including carbon cycle feedback processes that occur between land, ocean, and atmosphere. In the discussion, the highlighted seasonality-driven ocean-atmosphere CO₂ dynamics, particularly in the North Atlantic and Southern Ocean, offer valuable insights into

underexplored climate feedbacks related to seasonality. These feedbacks are particularly uncertain because of the lack of year-round temporal coverage of observed BCP variables. The amplification of seasonal pCO₂ outgassing in key ocean sink regions and its impact on long-term carbon sequestration is particularly noteworthy. The study highlights noteworthy spatial differences in the storage of anthropogenic carbon between biotic and abiotic simulations, particularly within specific ocean regions such as the subpolar Atlantic. The authors also identify the mechanisms driving these differences with a well-supported explanation, demonstrating how factors like the faster oceanic pCO₂ growth rate, higher initial surface Revelle factor, and seasonal variations in air-sea pCO₂ gradients contribute to the reduced carbon uptake in abiotic simulations.

We are grateful to reviewer #3 for this insightful comment and in acknowledging the uniqueness of our approach and importance contributions to the ocean carbon cycle community.

A few concerns could be addressed and do not prohibit publication. The decision to include only high- and low- end emissions scenarios raises questions about the exclusion of moderate scenarios, which could yield unique feedback patterns beyond averaging the extremes. Including a brief discussion of this choice would provide helpful context to the design of the study.

The two extreme scenarios (SSP1-2.6 and SSP5-8.5) were selected to illustrate the robustness of our conclusions that the absence of biological carbon pump would lead to reduced future land and ocean capacity to absorb excess anthropogenic CO₂ from the atmosphere across these emission pathways. In order to increase the comprehensiveness of our findings, we have decided to performed the additional SSP2-4.5 (moderate CO₂ emissions scenario) and have added both simulations with and without biology. The result is fully consistent and does not change our main conclusions. We have denoted this in the revised manuscript by including relevant information (on SSP2-4.5 results) throughout the text.

Key results of the paper rely on a direct link between increased/decreased primary productivity and increased/decreased carbon sequestration by the BCP. Increased primary productivity does not always lead to increases in effective carbon storage or even carbon export past shallow ocean layers. An acknowledgement of the nuanced relationship between primary productivity, carbon export, and sequestration in the discussion would be valuable. Variations in productivity by region or organism type may have differing impacts on BCP efficiency, complicating direct links between productivity reductions and CO₂ sequestration losses. Addressing this complexity would help the study in guiding future climate modeling efforts, carbon cycle research, and policy development.

We have included the following statement in the discussions section to acknowledge the described nuanced relationship and cite a reference:

L279-286: “In Earth system models, the BCP is generally positively associated with surface primary productivity, nevertheless there is a nuanced relationship between primary production, carbon export, and carbon sequestration. Spatial and temporal variations in surface productivity, and ecosystem structure, may have non-linear impacts on BCP efficiency, complicating the direct links between reductions in productivity and CO₂ sequestration (Frenger et al., 2024). Addressing this complexity would be valuable to guide future climate modeling efforts, carbon cycle research, and policy development.”

Several points in the main text could be clarified to strengthen the narrative.

Lines 38-40: The explanation of how the BCP influences the ratio of preformed to remineralized carbon, and its direct link to total ocean carbon storage, could be clearer. Consider stating directly that the BCP reduces surface preformed carbon by converting it into export production carbon, thus increasing remineralized carbon at depth. A clarification would help avoid ambiguity, emphasizing that the BCP primarily redistributes carbon within the ocean rather than storing it in an isolated reservoir. Consider rephrasing to clarify and emphasize the specific role of the BCP in partitioning ocean-atmosphere carbon stocks.

We have revised the sentence:

“The biological carbon pump (BCP) regulates the ratio of preformed to remineralised carbon dioxide dissolved in the ocean interior, and hence how much carbon is stored in the ocean.”

to:

L51-54: “Through mechanism (ii), the biological carbon pump (BCP) effectively reduces surface preformed DIC while increasing remineralised DIC at depth. This vertical redistribution and variations in preformed to remineralised DIC determines the partitioning of ocean-atmosphere carbon stocks.”

Lines 40-42: The reference to “removing remineralized carbon” was initially confusing without clearer context. Since remineralized carbon results directly from BCP activity, specifying whether the term refers to modeled carbon stocks or the conceptual absence of the BCP would reduce potential misunderstandings. This could be interpreted as the authors taking out remineralized carbon, so that the deep ocean has less carbon, and then re-equilibrating the model. Alternatively, this could mean that the BCP was “shut off”, and then they got a readjustment in a different way with a different amount of carbon.

The analytical relationships of Goodwin et al. (2008) consider removal of remineralized carbon from the ocean reservoir. We have revised the sentence:

“Using an atmosphere-ocean equilibrium relationship, it is estimated that the atmosphere CO₂ levels would be 150-240 ppm higher when the remineralised carbon is removed [5].”

to:

L57-59: “Using an atmosphere-ocean equilibrium relationship, it is estimated that when the remineralised carbon stock is removed, atmospheric CO₂ levels would be approximately 150-240 ppm higher in the new equilibrium state [5].”

Lines 43-45: Consider clarifying in particular what aspects of the carbon sink are considered to be dominated by the solubility pump rather than BCP– for example, its current magnitude and variability, or projected changes under climate scenarios?

We meant the projected long-term changes, i.e. increase, in carbon sinks. We revised the sentence:

“Currently the BCP are often considered to operate at steady state, with the physically-driven solubility pump considered to have dominated the current and future carbon sink and storage rates [6, 7], ...”

To:

L62-64: “Currently the BCP is often considered to operate at steady state, with the physically-driven solubility pump considered to have dominated the projected increase in current and future carbon sink and storage rates [6, 7], ...”

Lines 45-48: Consider mentioning the critical role of limited observational evidence and lack of consensus on current changes in BCP efficiency when discussing gaps in the understanding of the current and future BCP.

We revised the following:

“... partly because Earth system models project large uncertainties in future primary production [8] and partly because of a lack of knowledge on the non-linear interactions and feedbacks between BCP and other Earth system components.”

to:

L64-67: “... due to the limited observational evidence and knowledge on the non-linear interactions and feedbacks between BCP and other Earth system components. As a result, there is currently no consensus on the projected changes in BCP efficiency. ”

Lines 50-53: While the examples of modeling approaches used to study the BCP here are relevant, the list appears selective. Consider rephrasing to signal the listed examples are representative rather than exhaustive.

We have added “, among others.” at the end of the original sentence (L77-78).

Lines 53-56: The assertion that neglecting feedback processes leads to overestimation of the BCP's impact could benefit from a short explanation. Why does the omission consistently result in overestimation, rather than introducing more general uncertainty?

We agree that the net effect does not necessarily have to be an overestimation. We have rephrased the following:

“Such approaches can considerably overestimate the impact of changing BCP on the long-term ocean carbon uptake [9].”

to:

L80-81: “Such approaches can either underestimate or overestimate the impact of changing BCP on the long-term ocean carbon uptake [8].”

Lines 60-61: The term “marine ecosystem” might be too broad. If the study specifically shuts down the BCP, consider using that term to align more closely with the experiment design.

We replaced “marine ecosystem” with “BCP” (L85).